# Structure-Property Relationships in Suspension HVOF Nano-TiO$_2$ Coatings

**Feifei Zhang [1,2], Shuncai Wang [1,\*], Ben W. Robinson [2], Heidi L. de Villiers Lovelock [3] and Robert J.K. Wood [1]**

[1] National Centre for Advanced Tribology at Southampton (nCATS), School of Engineering, University of Southampton, Southampton SO17 1BJ, UK
[2] Surface Engineering, TWI Limited, Cambridge CB21 6AL, UK
[3] Oerlikon Metco WOKA GmbH, 36456 Barchfeld-Immelborn, Germany
\* Correspondence: wangs@soton.ac.uk; Tel.: +44-238-059-4638

**Abstract:** Hardness and tribological properties of microstructured coatings developed by conventional thermal spraying are significantly affected by the feedstock melting condition, however, their effect on the performance of nanostructured coatings by suspension high velocity oxy-fuel (HVOF) are inconclusive. In this work, nano-TiO$_2$ coatings with different degrees of melting (12%, 51%, 81%) of nanosized feedstock were deposited via suspension HVOF spraying, using suspensions with a solid content of 5 wt.%. All the coatings produced had dense structures without visible pores and cracks. Two TiO$_2$ crystal structures were identified in which the rutile content of the coatings increased with increased feedstock melting. Their mechanical, friction and wear behaviours largely relied on the extent of melting of the feedstock. The coating composed of mostly agglomerate particles (12% melted particles) had the lowest coefficient of friction and wear rate due to the formation of a smooth tribo-film on the wearing surface, while the coating composed of mostly fully melted splats (81% melted particles) presented the highest coefficient of friction and low wear rate, whose wear mechanism was dominated by abrasive wear and accompanied by the formation of cracks.

**Keywords:** HVOF; suspension; TiO$_2$; thermal spray; friction and wear behaviour

## 1. Introduction

In recent years, a modified thermal spraying process using a fine suspension of submicron or even nanostructured powders in a liquid phase as the feedstock material has gained an increased level of interest in the scientific world [1]. The use of suspensions opens up an entirely new class of spray materials for the production of nanostructured coatings by thermal spray technologies, as conventionally only tens of micron-sized powders can be used with a standard powder feeding device [2]. The introduction of the liquid phase in the suspension provides good flowability and therefore allows direct feeding of nanosized or submicron-sized feedstock, which enables the fabrication of finely structured coatings [1–3]. Suspension thermal sprayed coatings exhibit refined splats whose size is at least one order of magnitude smaller than that of the conventional thermal sprayed coatings [4], and the coating thickness can be controlled in a range from a few μm to several mm [1].

Thermal sprayed coatings are widely used in industry to improve the tribological properties in a number of applications, such as rolls, pump bodies and plungers, as well as machinery parts [3,5]. Recent studies have shown that nanostructured coatings exhibit outstanding properties, such as better sliding wear resistance than those of conventional ones [3,5–9]. For example, the wear rate of nanostructured Y$_2$O$_3$-ZrO$_2$ (YSZ) coatings, which had higher hardness, lay between 25% and 40% of that of conventional coatings [6,9]. It has also been observed that nanostructured coatings

exhibited enhanced crack propagation resistance against wear, even when they were not harder than the corresponding conventional coatings. For example, an improvement of three to four times of wear resistance under dry sliding conditions was observed for nanostructured $Al_2O_3$-13 wt.% $TiO_2$ coatings when compared with optimized microstructure coatings, even though they have a lower hardness [7,8]. Therefore, the overall hardness and wear resistance for nanostructured thermal sprayed coatings are not always correlated.

Generally, the hardness of materials is the most critical factor on wear resistance, although other factors, including ductility, toughness and microstructure, also play a role in the wear process [7,10]. However, there is still no established correlation between the amount of nanostructured zones embedded in the microstructure and the coating performance [3]. The tribological behaviour of the nanostructured coatings are more related to the amount of unmelted powder incorporated into the final coating [11]. The study on crack growth resistance of nanostructured thermal sprayed $Al_2O_3$-$TiO_2$ coatings showed that 60% or more of the crack arrest events are trapped within the partially-melted regions and deflected at the interface between partially-melted and fully-melted regions, compared to only 3%–12% in fully-melted splats [12]. To the extent of our knowledge, no detailed studies regarding the influence of melting conditions on the friction and wear behaviour of thermal sprayed nanostructured coatings have been published in the literature.

In the present study, three kinds of nanostructured $TiO_2$ coatings with different melting conditions of feedstocks were fabricated by varying the deposition parameters of the high velocity oxy-fuel (HVOF) process in order to ascertain how different coating structures affect the coating tribological performance. The constituent phases, microstructure, mechanical properties, friction and wear behaviour under dry sliding contact conditions of the coatings were examined in detail using a number of characterization techniques.

## 2. Materials and Methods

### 2.1. Suspension Preparation

A commercial $TiO_2$ nanopowder (Aeroxide P25, Degussa-Evonik, Hanau, Germany) was used as the $TiO_2$ feedstock. The $TiO_2$ suspensions were prepared in-house and consisted of 5 wt.% solid powder and 95 wt.% solvents. The solvents were mixtures of $H_2O$ and isopropanol (Table 1) and acted as a carrier during feedstock feeding.

**Table 1.** Deposition parameters of suspension high velocity oxy-fuel (HVOF) sprayed $TiO_2$ coatings.

| Label | Suspension Feed Rate, mL/min | Solvent, *v/v* | Spray Distance, mm | Fuel |
|-------|------------------------------|----------------|--------------------|------|
| S1 | 20 | $H_2O$:isopropanol = 9:1 | 130 | Propylene |
| S2 | 20 | $H_2O$:isopropanol = 10:0 | 100 | Hydrogen |
| S3 | 20 | $H_2O$:isopropanol = 9:1 | 150 | Hydrogen |

### 2.2. Spray Process

The as-prepared suspensions were deposited onto a commercially available stainless steel (AISI Grade 304) substrate (25 mm × 25 mm × 1.5 mm). All the samples were grit blasted with 100 mesh fused alumina abrasive prior to HVOF spraying. The coating deposition process was carried out at TWI Limited in Cambridge, UK, using a UTP Top Gun torch with a 22 mm long combustion chamber and a 135 mm long expansion nozzle mounted on an OTC AII-V20 robot. The suspension feed rate was controlled using an ISCO 260D syringe pump. The suspension was injected perpendicularly into the flame using a 0.3 mm nozzle mounted on the top of the torch at the combustion exit (Figure 1). Since the feedstock particle size used in suspension sprayed was much smaller compared with conventional spraying, there was much lower feedstock momentum and thermal inertia. The spray distance was reduced to compensate for the decreased particle kinetic energy. Thus, the heat flux from the flame

into the substrate was much higher, at least one order of magnitude more than the typical value for conventional spraying at similar conventional powder feed rates [13]. A water-cooling system was attached to the back of the substrates to extract heat and prevent distortion of the substrates by keeping the substrate temperature at a constant 55 °C. Compressed air was applied to further cool down the samples after spraying. A schematic of the suspension HVOF system is shown in Figure 1. The main suspension HVOF spraying parameters are listed in Table 2 [14,15].

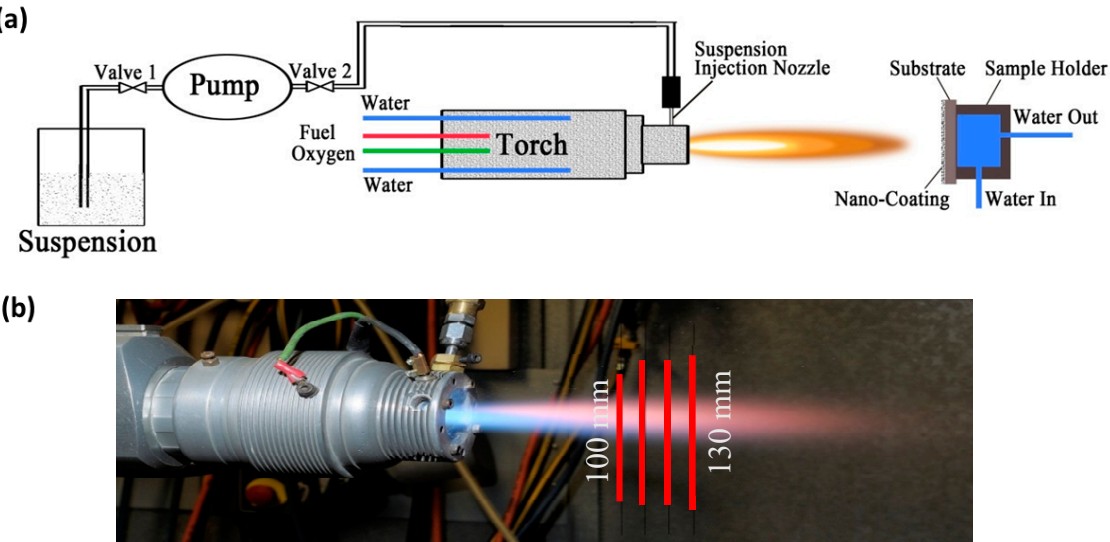

**Figure 1.** (**a**) Schematic of the suspension HVOF system for nanostructured $TiO_2$ coating. (**b**) Digital image of the HVOF propylene flame with suspension.

**Table 2.** Main suspension HVOF spraying parameters.

| Parameters | Value |
|---|---|
| Pass spacing, mm | 2 |
| Torch linear velocity, mm/s | 600 |
| Torch cooling system | Water cooling |
| Combustion chamber length, mm | 135 |
| Profile of suspension nozzle, mm | 0.3/orifice |
| Number of passes | 15 |
| Flame condition 1 | – |
| Propylene flow rate, slpm | 80.5 |
| Oxygen flow rate, slpm | 280.0 |
| Flame condition 2 | – |
| Hydrogen flow rate, slpm | 788.0 |
| Oxygen flow rate, slpm | 264.0 |

### 2.3. Coating Characterisation

The morphology of the samples (feedstock, surface, cross-section and worn scar of the coating) was characterised using scanning electron microscopy (SEM) (JEOL JSM 6500F, Tokyo, Japan). Cross-sectional samples were cold-mounted in resin, ground with SiC grit papers in stages from 120 down to 4000 mesh and finally polished with 0.4 μm $Al_2O_3$ slurry. All SEM images in the study were secondary electron images. A non-contact 3D optical profilometry (Alicona InfiniteFocus SL, Raaba/Graz, Austria) was used to characterise the surface roughness of the as-deposited specimens (20×). At least three readings were taken for each sample and the average value was recorded.

The crystalline phases of the coatings were determined by X-ray diffraction (XRD) using a Bruker D2 PHASER diffractometer (Billerica, MA, USA) in the reflection mode with Cu-Kα radiation ($\lambda$ = 0.154 nm). The scan step was 0.02°, with a step time of 0.5 s in the 20–80° 2θ range. Peaks of

phases were analysed by Jade XRD software (version 5.0). Titanium oxide has two structures—anatase and rutile. Both anatase and rutile are tetragonal structures with different *c/a* ratios of 2.52 and 0.64, respectively, and different densities of 3.915 and 4.276 g/cm$^3$, respectively. Anatase has a relative larger spacing which may accompany with easy slide. The volume percentage of rutile ($C_R$) was determined according to the following equation [16]:

$$C_R = \frac{13\,I_R}{8I_A + 13\,I_R} \tag{1}$$

where $I_A$ and $I_R$ are the X-ray intensities of the anatase (101) and the rutile (110) peaks, respectively.

Nano-indentation testing was performed by NanoTest Vantage (Micro Materials Ltd., Wrexham, UK) onto a polished cross-sectional surface of suspension HVOF TiO$_2$ coating using a three-sided pyramidal Berkovich diamond indenter tip with a diameter of 200 nm. A fixed penetration depth of 300 nm, loading rate of 0.4 mN/s and holding time of 40 s were used for testing. Multiple indentations separated by 6 μm were performed for each load value. The hardness and the elastic modulus were recorded by the nano-indentation software, and the Poisson's ratio was assumed to be 0.27 [17]. Five readings were taken for the final results.

Reciprocating wear testing was carried out on the surface of as-deposited coatings TE77 (Phoenix, UK) to determine the friction and wear behaviour of solid surfaces in sliding contact. All the tests were performed under dry sliding conditions under a constant load of 5 N (the initial Hertzian contact pressure was 1.27 GPa), at an ambient temperature of 23 ± 1 °C and 60 ± 1% relative humidity (ASTM G133–05) [18]. The stroke length was 2.69 mm with a sliding frequency of 1 Hz (the average sliding speed was 5.38 mm/s), and the total sliding distance was 3.228 m. A sintered Al$_2$O$_3$ ball (manufacturer's nominal hardness of 19 GPa) with diameter of 6 mm was used as the counter body. A piezo electric transducer was used to measure the friction force. The coefficient of friction and the sliding time were recorded automatically during the test. At the end of the test, the sample was cleaned by compressed air flow to remove loose debris. The track profile was acquired by a non-contacting 3D microscope (Alicona InfiniteFocus SL, Raaba/Graz, Austria), and at least five profile measurements were taken for each wear track. The corresponding specific wear rate was calculated from the equation:

$$K = \frac{V}{SF} \tag{2}$$

where $V$ is the wear volume in mm$^3$, $S$ is the total sliding distance in metres and $F$ is the normal load in newton.

## 3. Results

### 3.1. Microstructure

The prepared suspension was dried and the particles were observed by SEM (Figure 2a). The nanoparticles were distributed uniformly within the suspension, with small agglomerates only about several hundreds of nanometres in size (circled in Figure 2a). No stratification could be observed in the sedimentation test even after the TiO$_2$ suspension had been left untouched for 36 h (inset in Figure 2a). In order to further characterise the stability of the 5 wt.% TiO$_2$ suspension, the zeta potential was monitored over a period of 120 min (Figure 2b). When the zeta potential is higher than 30 mV or lower than −30 mV, it is accepted that the suspension can resist strong agglomeration and can be electrically stabilized [19]. During the first 60 min, the electrokinetic potential varied between −41 and −57 mV, and then tended to stay stable with little variation between −49 and −56 mV. This was because the electrical charge at the double layer of particles in freshly prepared suspensions is not electrically stabilized [20] and the equilibrium state between particles and dispersed solution can be reached in 1 h for suspensions with a solid content of 5 wt.%.

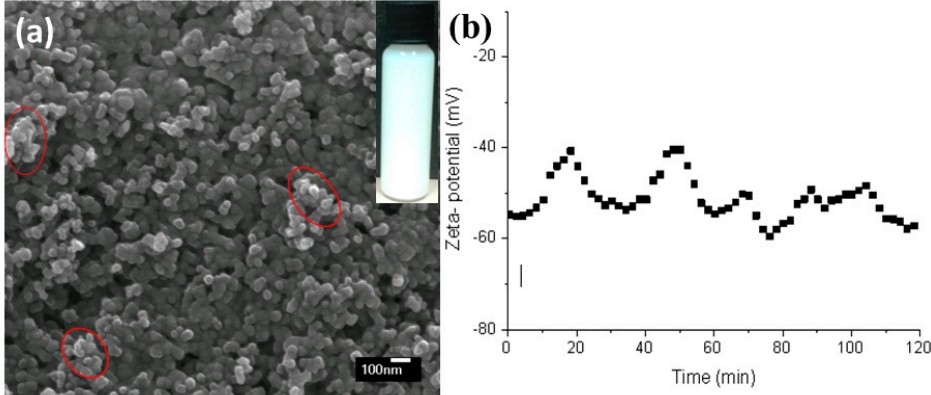

**Figure 2.** (**a**) Scanning electron microscopy (SEM) image of TiO$_2$ nanoparticles in the suspension (the inset is the digital image of the sedimentation test of suspension after 7 days); (**b**) zeta potential changes with time.

Figure 3 shows the surface and cross-section microstructures of suspension HVOF TiO$_2$ coatings deposited using different parameters. Typical surface features include re-solidified particles, semi-melted, fully-melted and/or agglomerated particles caused by different thermal histories of each individual particle/agglomerate which then lead to different impacting behaviours onto the substrate [14]. The as-deposited coatings generally have very low surface roughness and exhibit smooth surface features compared with that of conventional thermal sprayed coatings [21]. The average surface roughness ($R_a$) for the three coatings (as shown in Table 3) ranged from 0.53 to 1.18 µm. The cross-sections of suspension HVOF coatings clearly show that they possessed "bimodal" distributed microstructures, where fully-melted zones (bright zones) correspond to the rutile phase and agglomerate zones (grey zones) correspond to the anatase phase, based on Raman spectra analysis as reported by in [22]. Furthermore, the suspension HVOF TiO$_2$ coatings exhibited a dense structure, with no visible pores and no obvious defects when observed by SEM. The coatings adhered well to the substrate along the surface profile. None of the coatings appeared to have the typical micro-cracks frequently observed in conventional thermal sprayed coatings, with these being the result of the relaxation of thermal stresses generated during processing associated with significant convective heat input [23]. The coating thicknesses were 5.6 ± 1.7, 15.5 ± 2.1 and 8.1 ± 2.8 µm for coatings S1, S2 and S3, respectively.

The cross-sectional images were converted from greyscale images into binary images using a threshold to evaluate their extents of melting, as shown in Figure 4. The extent of feedstock melting was ca. 12%, 51% and 81% for coatings S1, S2 and S3, respectively. The crystalline structure of the coating was analysed by X-ray diffraction patterns, as shown in Figure 5. The P25 nanopowder was composed of anatase and rutile phases in the proportion 81:19. No other phases were observed in the coatings compared with P25 feedstock, except the peaks at around 44.5° that revealed the austenite phase of the stainless steel substrate due to the low thickness of the coating. An increase of rutile peak intensity was observed in the XRD patterns for the coatings compared with P25 powder. This indicated that anatase-to-rutile phase transformation occurred during the deposition process because of heat transfer from the HVOF flame jet. The rutile content ($C_R$) was 23.6%, 63.0% and 70.0% for coatings S1, S2 and S3, respectively, a tendency consistent with the extent of feedstock melting observed by SEM. More fully-melted regions in the coating led to higher $C_R$, which is consistent with results in [22].

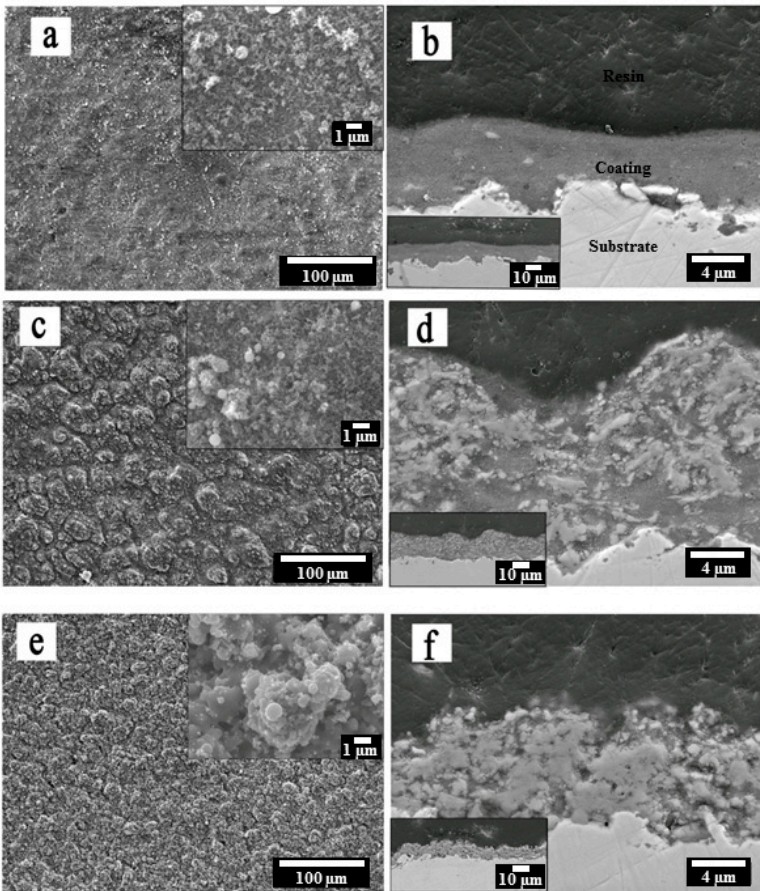

**Figure 3.** SEM images of surface and cross-sectional coating morphologies: (**a**,**b**) S1, (**c**,**d**) S2 and (**e**,**f**) S3.

**Table 3.** Values of the coefficient of friction and the specific wear rate of different samples.

| Samples | Hardness $H_v$, GPa | Surface Roughness $R_a$, μm | Wear Rate, $\times 10^{-7}$ mm$^3$/Nm | Coefficient of Friction |
|---------|---------------------|------------------------------|----------------------------------------|--------------------------|
| 304SS   | 4.7 ± 0.3           | 0.56 ± 0.14                  | 5.13 ± 0.04                            | 0.55 ± 0.05              |
| S1      | 2.1 ± 0.3           | 0.53 ± 0.14                  | 0.83 ± 0.03                            | 0.35 ± 0.02              |
| S2      | 4.0 ± 0.9           | 1.18 ± 0.18                  | 5.13 ± 0.13                            | 0.48 ± 0.04              |
| S3      | 7.8 ± 0.4           | 0.96 ± 0.17                  | 1.77 ± 0.05                            | 0.62 ± 0.03              |

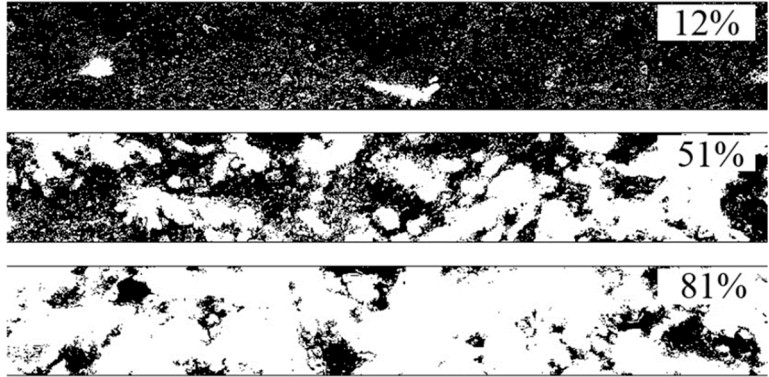

**Figure 4.** Binary images of suspension HVOF TiO$_2$ coatings converted from their greyscale images by image thresholding in order to assess the melting percentages: S1 (12%), S2 (51%) and S3 (81%).

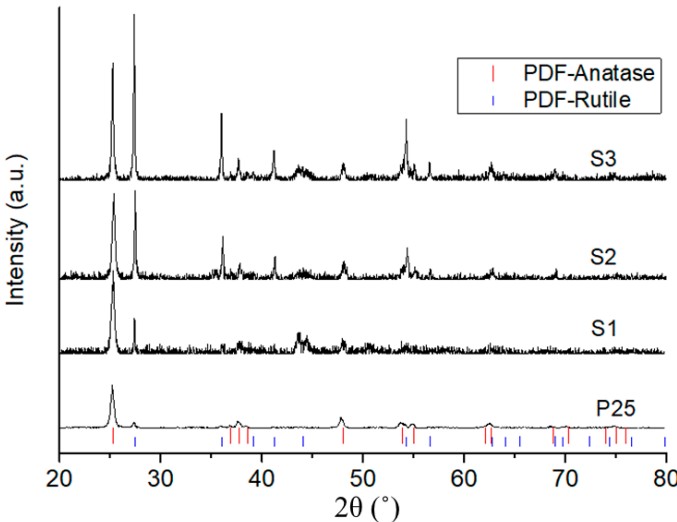

**Figure 5.** X-ray diffraction (XRD) analysis of suspension HVOF TiO$_2$ coatings.

## 3.2. Mechanical Properties

The hardness and elastic modulus in different melting zones of the coating cross-section were evaluated by nano-indentation with a depth of 300 nm. Hardness is an indicator of the irreversible or plastic deformation behaviour of the test material [24]. The average hardness values for the agglomerated zone (mostly the anatase phase) and fully-melted zone (mostly rutile) were 2.1 and 7.8 GPa, respectively (Figure 6). These values are lower than those reported in the literature for suspension thermal sprayed TiO$_2$ coatings, which were above 8.5 GPa [25,26]. This can be ascribed to the lower organic content of the suspension and the lower suspension feed rate used in this study, with both leading to less melting of the feedstock. For example, some of the TiO$_2$ splats had a fully-melted solid core with a partially-melted or agglomerated interior structure (as circled in Figure 7) and the loosely accumulated splats tended to build more pores. Hardness measurements have been found to be sensitive to these built flaws generated in the coating, particularly porosity [27]. The elastic modulus of various zones for suspension HVOF TiO$_2$ coating lay between 27% and 50% of that of the bulk TiO$_2$ (282.0 GPa), and the elastic modulus of the fully-melted zone (135 GPa) was only slightly lower than that of a coating produced using a conventional HVOF process (164 GPa) [26]. This was within the expected range (20%–50%) for thermal sprayed coatings when compared with bulk material [28,29].

The variations of both hardness (*H*) and elastic modulus (*E*) across different melted zones show that they are closely related to the coating microstructure. Both of them increased as the amount of fully-melted particles increased (Figure 6). The fully-melted zone had the highest hardness and elastic modulus values due to its lower porosity. The lower values of elastic modulus and hardness compared to that of equivalent bulk materials can be attributed to the unique microstructure of suspension thermal sprayed deposits, which is typically composed of fully-melted, partially-melted and agglomerated particles.

The extent of the recovery during nano-indentation depends on the hardness-to-elastic modulus ratio (*H/E*). The *H/E* is an indicator of a material's capacity to absorb or dissipate energy, which is lower for more plastic materials and becomes higher for more elastic materials. The fully melted zones had the highest elasticity index and showed superior elastic property, while the agglomerated zones were more plastic (e.g., Figures 6 and 7). SEM examination of indents on a number of melted zones revealed that there were no cracks present around the indents (Figure 7). All the indentation areas were smooth, without any sign of incipient crack formation or material accumulation. A shallow and small indentation impression was observed for fully melted zones, compared with much larger ones for agglomerated zones (e.g., Figure 7a,c).

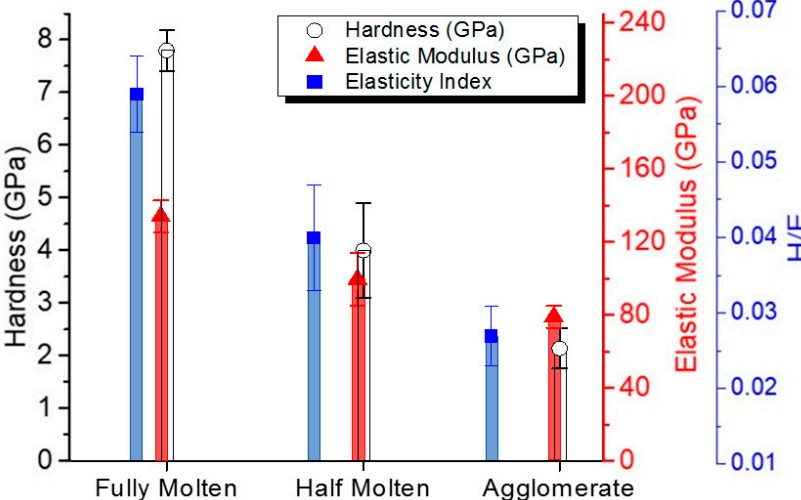

**Figure 6.** Comparison of nano-indentation results (*H*, *E* and *H/E*) on different zones of the suspension HVOF TiO$_2$ coating (S2).

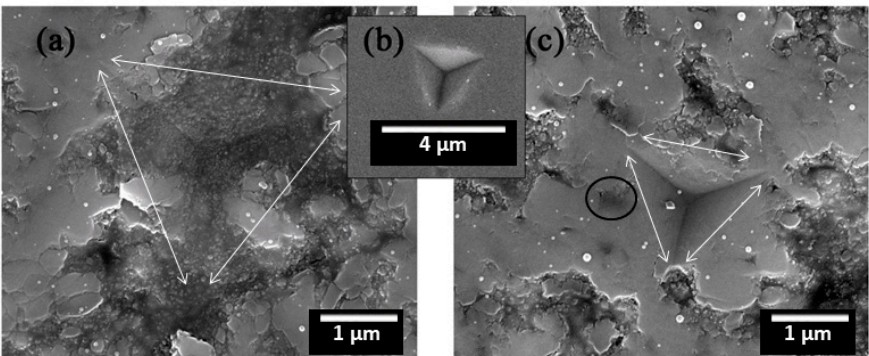

**Figure 7.** Morphologies of nano-indents: (**a**) region of agglomerated particles, (**b**) substrate-304 SS and (**c**) region of fully melted splats.

*3.3. Friction and Wear Behaviour*

As the thickness of the deposited coatings was very thin (less than 20 μm, compared with several hundreds of micrometres for conventional thermal sprayed coatings), the reciprocating wear test only ran for 600 s to avoid the effect of the substrate. The as-sprayed surface roughness did not seem to be critical for wear and friction performance of the developed coatings in the study when their $R_a$ varied between 0.53 and 1.18 μm (Table 3). Suspension HVOF TiO$_2$ coatings prepared in this study had various wear rates, in the range of 0.83–5.13 × 10$^{-7}$ mm$^3$/Nm. The coefficient of friction varied from 0.35 to 0.62, which is lower than that of conventional plasma sprayed and suspension HVOF TiO$_2$ coatings obtained from dry sliding ball-on-disc tests using the same counterpart 6 mm Al$_2$O$_3$ ball but with a load of 2 N as found in the literature, i.e., ca. 0.90 [30]. With the increase of the melting extent, the friction coefficients increased though the wear resistance varied differently. The coefficient of friction curves of different samples (Figure 8) indicated that all the suspension HVOF TiO$_2$ coatings had smoother curves compared with that of 304SS, regardless of the extent of the melting of the feedstock. However, the coatings with different melting conditions of feedstock had very different coefficient of friction values. The coating with mostly agglomerated particles (S1) showed the lowest coefficient of friction, and the coating with mostly fully melted particles (S3) presented the highest coefficient of friction. It can be assumed that coefficient of friction is related with particle melting extent, which is then reflected by phase structure, i.e., anatase has a lower coefficient of friction than rutile.

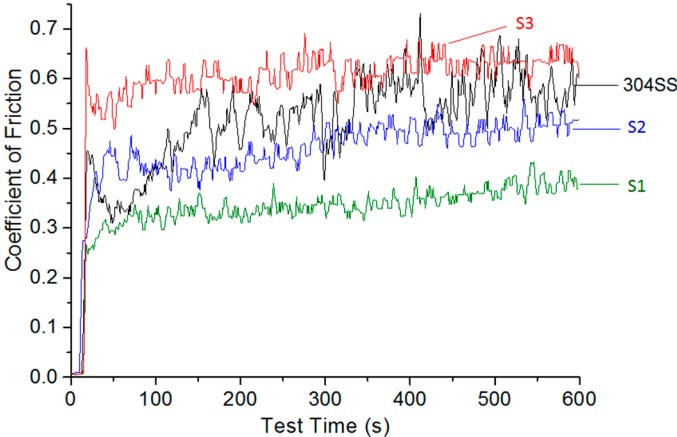

**Figure 8.** Evolution of the coefficient of friction curves of suspension HVOF TiO$_2$ coatings.

In order to further study the wear mechanism, the surface morphologies of the wear scars were examined by SEM. Uncoated 304SS substrate had a specific wear rate of $5.13 \pm 0.04 \times 10^{-7}$ mm$^3$/Nm (Table 3). Its coefficient of friction varied between 0.38 and 0.72 (Figure 8), consistent with the typical values for the contact between ceramic and metallic materials [31]. The large fluctuations in the data were typical for poor tribological materials with significant adhesive wear and stick-slip tendencies. Wear scar analysis confirmed that a large amount of wear debris from the substrate adhered to the contact surfaces of both the substrate and the counterpart Al$_2$O$_3$ ball (Figure 9). Severe plastic deformation of steel wear debris and the generation of a series of grooves on the wear scar surface indicated that two-body ploughing also occurred during the wear process (Figure 9a,b). The presence of a high amount of oxygen (overlap with Cr peak) on the 304SS surface (compared with EDS analysis of bare 304SS substrate in Figure 9d) indicated strong oxidation due to elevated temperatures between the contact surfaces (Figure 9c). The wear process of uncoated 304SS substrate was dominated by adhesive wear and two-body ploughing and had a typically fluctuating coefficient of friction curve.

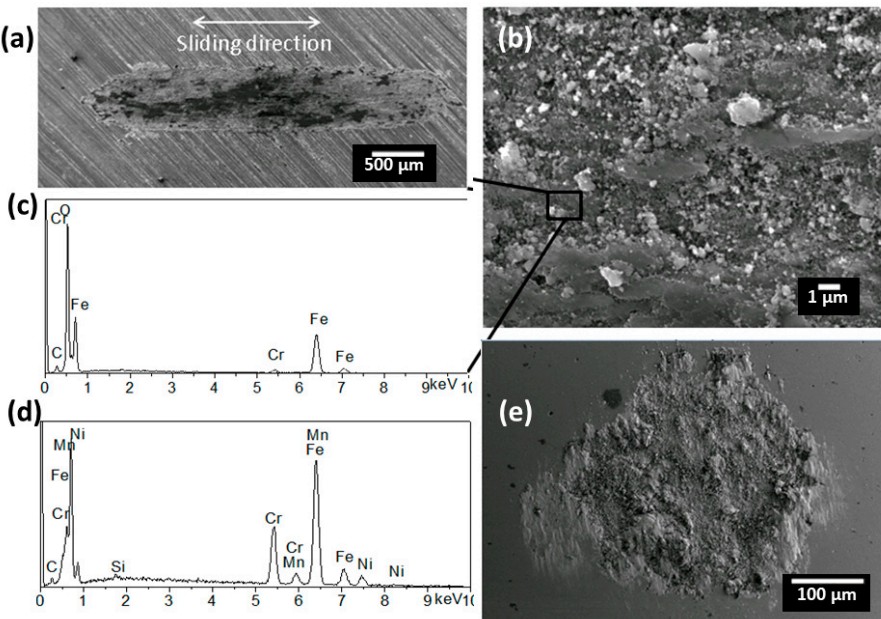

**Figure 9.** Wear track of 304SS under (**a**) low magnification and (**b**) high magnification; (**c**) EDS analysis of area in image (**b**); (**d**) area EDS analysis of as-received bare 304SS substrate; (**e**) wear scar of counterpart Al$_2$O$_3$ ball.

It is widely considered within the thermal spray community that a higher amount of fully melted splats is essential for improving the strength of thermal spray coatings [32] and, as a result, coatings composed of mostly agglomerate particles have received much less attention, especially for tribological applications. In this study, coating S1, with a feedstock melting percentage of 12% ($C_R$ = 23.6%), showed a very low coefficient of friction and low wear rate (Figures 8 and 10). High magnification examination of the coating in the cross-section showed that it was mainly composed of agglomerated spherical granular $TiO_2$ particles (Figure 10a). From the SEM analysis, the wear scar was small and had a smooth surface without wear debris (Figure 10b,c). Only a few small localized cracks shorter than 3 μm (as observed under SEM) and perpendicular to the sliding direction could be seen on the worn surface (Figure 10c). It is assumed that the existence of agglomerated granular particles plays an important role in arresting and deflecting crack propagation during wear. The act of applying the test load led to a densification of the coating on the worn surface and a layer of compact tribo-film was therefore formed by the deformation of splats, which was mainly $TiO_2$ (Figure 10a,c), which provided a low coefficient of friction. The EDS analysis revealed the exposure of substrate material (iron) as the coating was worn through (Figure 10d), indicating that the tribo-film can be continuously formed and worn off because of the much higher hardness of the $Al_2O_3$ ball compared with that of the coating. Only very little $TiO_2$ debris was found adhering onto the surface of the $Al_2O_3$ ball because of the formation of tribo-film (Figure 10e). Therefore, the improved friction and wear properties of the coating composed of mostly agglomerated feedstock can be ascribed to the effective hindrance of crack generation and propagation of the granular feedstock, which leads to enhanced integral deformation resistance of the nanostructured coatings.

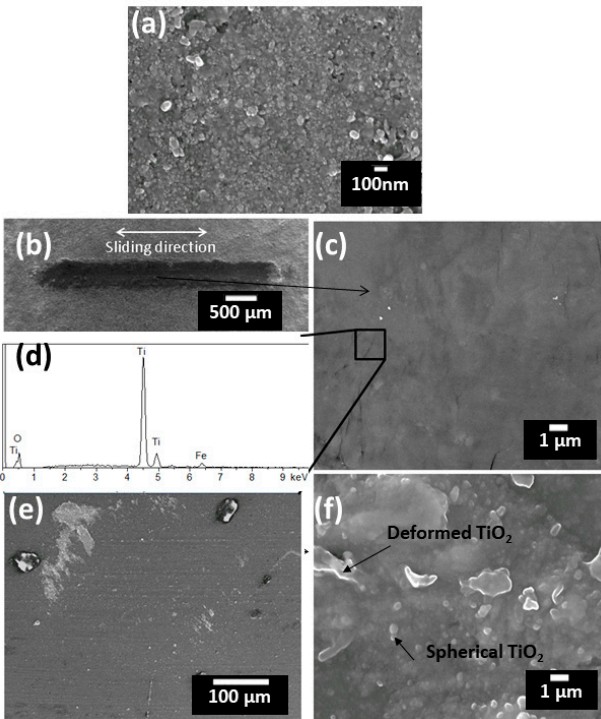

**Figure 10.** Coating S1: (**a**) cross-sectional SEM view of coating S1, wear track under (**b**) low magnification and (**c**) high magnification; (**d**) EDS analysis of area in image; wear scar of counterpart $Al_2O_3$ pin under (**e**) low magnification and (**f**) high magnification.

With increased extents of feedstock melting, the coating S2 had a higher coefficient of friction and complex wear behaviour (Figures 8 and 11). Unlike the previous coating (S1) composed of mostly agglomerated particles, a considerable amount of pulled-out debris was found on the worn surface of S2, especially along the edges of the wear scar (Figure 11a). From the enlarged image of the wear

scar, the generated cracks were much bigger compared with that of coating S1, which was due to the increased level of the extent of melting and the corresponding decrease in plasticity (Figure 11b). The inhibition and deflection effects of preserved nanostructured zones on the crack propagation were clearly observable, confirming that the existence of nano-zones can act as crack arresters (i.e., energy absorbers) and improve the toughness of the coating. A similar phenomena was described in [33] by applying a micro-indentation test on a nanostructured HVOF $TiO_2$ coating. The EDS analysis showed that the worn surface of the coating was mostly $TiO_2$ with a small amount of iron from the substrate, whilst no aluminium, which would indicate counterpart alumina material transfer, could be detected (Figure 11c). Fine and irregular wear debris composed of sintered big agglomerates and fully-melted splats were observed on the contact surfaces (Figure 11d,e). Since a coating that is composed of half-melted feedstock will have an increased level of build defects such as weak intersplat boundaries, the vertically applied load can lead to the displacement of loosely attached $TiO_2$ particles on the coating surface asperities. These fragments can be transferred to the edges of the wear scar or remain at the interface and act as a third body. The trapped debris between the contact bodies accelerates particle exfoliation and the delamination of the tribo-film during the reciprocating sliding wear process. More wear debris could be formed as a resulted of surface fatigue, brittle fracture and crumbling. The specific wear rate of coatings with moderate extents of feedstock melting can therefore be very high when such coatings present high levels of defects.

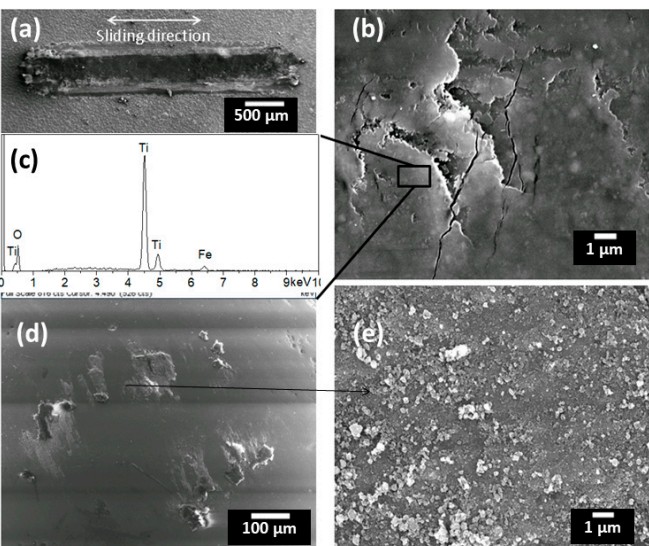

**Figure 11.** Wear track of coating S2 under (**a**) low magnification and (**b**) high magnification; (**c**) EDS analysis of area in image (**b**); wear scar of counterpart $Al_2O_3$ pin under (**d**) low magnification and (**e**) high magnification.

For coating S3 that was composed of mostly fully melted feedstock, its coefficient of friction was higher than the previous two types of coating (Figures 8 and 12). From Figure 12a, the wear scar was clean and no obvious wear debris had accumulated on the surface. It can be clearly seen from the enlarged image of the wear track surface that the fatigue crack propagation runs perpendicular to the sliding direction (Figure 12b), which was caused by surface tensile stresses. From Section 3.2, the overall hardness of the coating increased with the increase of the number of fully melted splats, and at the same time the plasticity decreased. Furthermore, the increase of the proportion of fully melted splats caused a loss of nano-features in the coating, which restrained the formation of the tribo-film. These led to a decrease in crack propagation resistance and a correspondingly high coefficient of friction. The presence of aluminium in the EDS spectrum on the worn surface indicates that material transfer occurred between the coating and the $Al_2O_3$ ball surface (Figure 12c). Further investigation of the surface of the counterpart $Al_2O_3$ ball shows that many detached fragments were adhered to the

edge of the wear track and the ball surface had experienced significant abrasion in the centre of the wear scar (Figure 12d,e). In summary, the coating composed of mostly fully-melted feedstock still had good wear resistance due to its high hardness, albeit with a higher coefficient of friction (Table 3).

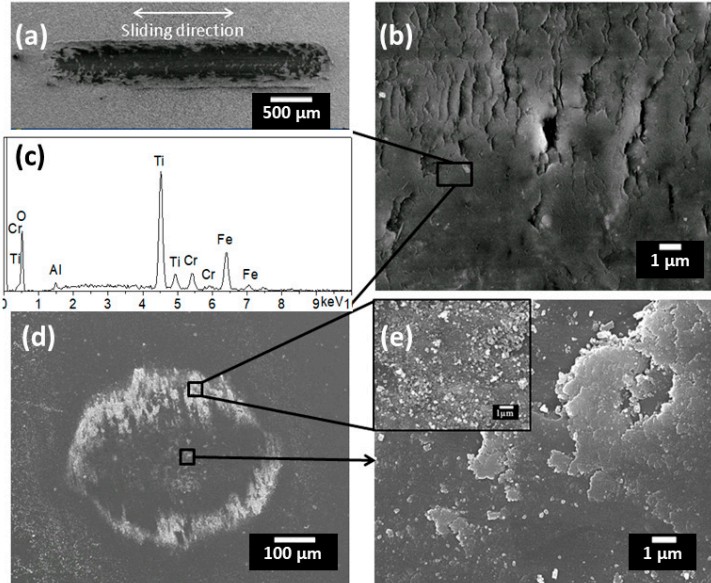

**Figure 12.** Wear track of coating S3 under (**a**) low magnification and (**b**) high magnification; (**c**) EDS analysis of area in image (**b**); wear scar of counterpart Al$_2$O$_3$ pin under (**d**) low magnification and (**e**) high magnification.

## 4. Discussion

From the information above, the friction and wear responses of suspension HVOF TiO$_2$ coatings are difficult to predict by adopting the principles of wear behaviour that are often used for conventional engineering materials. An increase in hardness is often associated with increased resistance in sliding, low-angle erosive or abrasive wear conditions for conventional materials within a particular material category [2,34], but it does not always improve the wear resistance of thermal sprayed coatings because of the complexity of their structure, which contains different microstructural features, including pores, cracks and fragments [5,35]. The existence of cracks and horizontal pores are ideal "shear faults" for plastic deformation and are favourable crack initiation points [36]. During the wear process, the vertically applied load can cause displacement of deposited splats that are mechanically adhered to each other. This debris trapped between the contact surfaces causes three-body abrasion and thus aggravates the wear of the coating. Nano-indentation demonstrated that the properties (hardness, elastic modulus, and elastic index) for the suspension HVOF TiO$_2$ coating were dissimilar for zones with different extents of melting of the feedstock (Figure 6). The more melted microstructure contained more rutile and had higher hardness, while the surrounded agglomerated zone consisted mostly of anatase and was softer [25]. Higher hardness and stiffness leads to a more brittle nature of the coatings that have more melting [37], whilst the higher induced shear force during wear causes a higher coefficient of friction. From wear tests, when the coating had a negligible level of built defects, the fully melted splats improved the cohesion strength and hardness, which thus led to good wear resistance, and the agglomerated zone was beneficial in getting a low coefficient of friction due to its good plasticity. In general, the wear mechanism of suspension HVOF TiO$_2$ coatings always involved the formation of a smooth and dense surface film (tribo-film) on the surface of the wear track and its progressive delamination and removal caused by wear debris and surface fatigue during wear (Figure 11). The observed results for suspension HVOF TiO$_2$ coatings in the study prove that the increase in friction appears to be related to the extent of melting of feedstock, i.e., increased levels of the rutile phase.

A further experiment has been carried out to confirm the relationship between coefficient of friction and rutile content. A series of coatings were deposited using different spray distances with propylene as fuel and a suspension feed rate of 20 mL/min. When increasing spray distances from 100 mm, the impacting speed of nanoparticles onto the substrate and flame temperature would become lower (Figure 1b). Coating microstructures, as shown in Figure 13, indicates different levels of feedstock melting conditions under each spray distance, and the extent of melting decreases when spray distance increases. The $C_R$, wear rates and friction coefficients are listed in Table 4. These results provide evidence that the agglomerated zone (mainly anatase) is beneficial for a low coefficient of friction and improved wear resistance. This could be explained by the easy separation of the anatase tribofilm along its spacious c-axis. However, further work is needed to verify this.

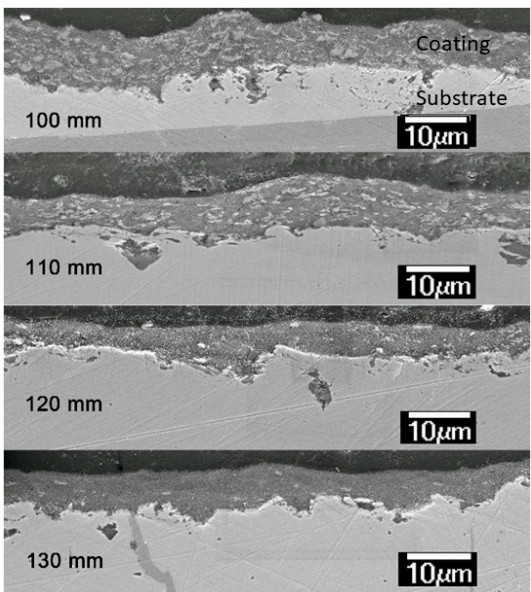

**Figure 13.** Influence of spray distance on the coating microstructure.

**Table 4.** List of properties of coatings with different $C_R$ ($H_2O$:isopropanol = 9:1).

| Spray Distance | $C_R$, % | Surface Roughness, μm | Coefficient of Friction | Specific Wear Rate, $\times 10^{-7}$ mm$^3$/(N m) |
|---|---|---|---|---|
| 100 mm | 58 | 0.72 ± 0.10 | 0.68 ± 0.04 | 2.47 ± 0.07 |
| 110 mm | 53 | 0.44 ± 0.15 | 0.54 ± 0.05 | 2.01 ± 0.09 |
| 120 mm | 45 | 0.67 ± 0.08 | 0.38 ± 0.03 | 0.64 ± 0.07 |
| 130 mm | 41 | 0.53 ± 0.14 | 0.36 ± 0.02 | 0.83 ± 0.02 |

## 5. Conclusions

Nanostructured $TiO_2$ coatings were deposited onto a stainless steel substrate (304SS) by controlling the level of heat input to feedstocks through various combinations of suspension HVOF spray parameters. Three types of coatings with different particle melting conditions (12%, 51%, 81%) were successfully prepared with dense structures and without visible pores and cracks.

The following conclusions allow us to better understand how the existence of different nanostructures affects the coating properties, providing guidance on coating design for different applications:

- Increasing extents of feedstock melting corresponded to increased rutile contents in the coatings, which led to an increase in overall hardness with a reduced plasticity.
- The as-sprayed surface roughness did not seem play an important role for tribological performance of the developed coatings when their Ra varied between 0.53 and 1.18 μm.

- The coating composed of most agglomerate particles (12% melted particles) had the lowest coefficient of friction, whereas the coating composed of mostly melted particles (81% melted particles) presented the highest coefficient of friction. Results also indicate that a higher fraction of agglomerated particles (proportional to anatase content) were beneficial to the formation of tribo-film at sliding surfaces.
- Wear resistance of the coatings were proven to be not rational to their hardness. The coating with mostly agglomerate particles (12% melted splats) had the lowest wear rate and the coating with moderate melted particles (51%) had the worst performance against wear.

**Author Contributions:** Conceptualization, F.Z., S.W., H.L.d.V.L. and R.J.K.W.; Methodology F.Z., S.W., B.W.R. and H.L.d.V.L.; Validation, S.W., H.L.d.V.L. and R.J.K.W.; Formal Analysis, F.Z.; Investigation, S.W.; Resources, B.W.R. and H.L.d.V.L.; Data Curation, F.Z. and S.W.; Writing—Original Draft Preparation, F.Z.; Writing—Review and Editing, S.W., H.L.d.V.L. and R.J.K.W.; Visualization, F.Z.; Supervision, S.W., H.L.d.V.L. and R.J.K.W.; Project Administration, S.W. and H.L.d.V.L.

**Funding:** This work was supported by University of Southampton, the TWI Limited (Cambridge, UK) and the China Scholarship Council (CSC).

**Acknowledgments:** The authors would like to thank Andrew K. Tabecki from the Surface Engineering section at TWI for his assistance with coating deposition.

**Conflicts of Interest:** The authors declare no conflict of interest.

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
