# Peer review of "Structure-Property Relationships in Suspension HVOF Nano-TiO2 Coatings"

_coatings, doi:10.3390/coatings9080504_

Round 1

Reviewer 1 Report

Materials and Methods:
Line 79:     The link to Table 1 in the manuscript is misleading.
Line 89:     Is it necessary to cite this source? The above information on the coating process is sufficient. A reference to Table 2 would be desirable.
Line 123+:     How many wear tracks were examined per coating?

Results:        
Line 166:    A consideration of deposition rate would be useful.
Line 186:    In order to assess the suitability of binary images for determining the content of rutile and anatase, a cross comparison with grayscale images would be helpful. Because of the XRD measurements, different values exist for the percentages of rutile and anatase respectively.
Line 238:    For which sample the maximum value of 17.45*10-7 mm³/Nm was determined?
Line 250:    Is it correct that the S2 coating has the same wear rate as the substrate?
Line 377:    What is the significance of the decreasing fraction of rutile when increasing the spraying distance with regard to the phase transformation of anatase --> rutile?
The serious difference in wear resistance between 110 mm and 120 mm spraying distance with a minimal change in the anatase / rutile ratio is not conclusive.

Conclusions:
To what extent can the influence of as-sprayed roughness on friction and wear behaviour be excluded?
How does the author define the term tribofilm? Can the formation of a tribological film be proven by the wear tracks?

Author Response

We are grateful to the reviewer whose comments are very helpful for us to improve the quality of the current research paper. Please find the individual reply (blue) and the corresponding text highlighted yellow in the manuscript.

Materials and Methods:
Line 79:     The link to Table 1 in the manuscript is misleading.
Reply: Thanks. We have added an extra column in Table 1.

Line 89:     Is it necessary to cite this source? The above information on the coating process is sufficient. A reference to Table 2 would be desirable.
Reply: Thanks. We moved this reference at end of section 2.2, and add the reference 15 for Table 2.

Line 123+:     How many wear tracks were examined per coating?
Reply: Five measurements which were added in line 129.

Results:        
Line 166:    A consideration of deposition rate would be useful.

Reply: Thanks. The awareness of the deposition rate is helpful but in this research was focussed on the melting extend (varied ratios between anatase and rutile.

Line 186:    In order to assess the suitability of binary images for determining the content of rutile and anatase, a cross comparison with grayscale images would be helpful. Because of the XRD measurements, different values exist for the percentages of rutile and anatase respectively.

Reply: Thank you for your suggestion. However, it is difficult to compare the values obtained by binary images and XRD measurement: the binary images converted from SEM images show contract from the unmelts or melts with different phase ratios, while XRD provide the precise crystalline percentage.

Line 238:    For which sample the maximum value of 17.45*10-7 mm³/Nm was determined?

Reply: Sorry for this mistake. Wear rate for coatings involved in this study varies between 0.83-5.13*10-7 mm³/Nm, text has been changed in line 238. The value of 17.45*10-7 mm³/Nm is the wear rate of a high defect coating we prepared in the PhD project, which is not included in this paper.

Line 250:    Is it correct that the S2 coating has the same wear rate as the substrate?

Reply: Thanks for questioning. We did re-check reciprocating wear tests on the substrate to confirm this result. The wear rate was calculated by results from five profile measurements on wear scars.

Line 377:    What is the significance of the decreasing fraction of rutile when increasing the spraying distance with regard to the phase transformation of anatase --> rutile?
The serious difference in wear resistance between 110 mm and 120 mm spraying distance with a minimal change in the anatase / rutile ratio is not conclusive.

Reply: The spray distance can alter spray particle velocity and temperature then affect the phase transformation from anatase to rutile. In this study as shown in Figure 1b, the spray distance of 100 mm lies in the hottest and fastest zone of the flame. With the further increase of spray distances, the phase transformation were reduced, which accompanied with decrease of friction coefficients.

Regarding the abnormal rutile content at 120 mm, by comparing the phase contrast in the SEM images of Fig. 13, we corrected the CR% value to 45 in Table 4.

Conclusions:
To what extent can the influence of as-sprayed roughness on friction and wear behaviour be excluded?
Reply: In this study, as-sprayed surface roughness does not seem to be critical for developed nanostructured coatings developed when their Ra varies between 0.53 and 1.18 µm. Text has been added in line 225 and line 378.

How does the author define the term tribofilm? Can the formation of a tribological film be proven by the wear tracks?

Reply: The tribofilm in this study was TiO2 either with anatase or rutile structures. The statement was added in line 266.

Reviewer 2 Report

In this paper nanostructured TiO2 coatings after depositing onto a stainless steel substrate using the suspension HVOF spray process were described. Different feedstock melting conditions were used and microstructures, hardness, mechanical properties as well as friction and wear behavior were tested. Some remarks can be done:

Key words are incomplete. It  would have been better “friction and wear behavior” or “tribological  behavior”.

What software was used to fulfill XRD phase’s analysis?

The bar graph should be used in Figure 6 instead of the point graph.

In 3.3 section the specific wear rate is analyzed (see 135 line), but not wear rate.

Figure 9 is too small; its components should be increased; spectrum 1 and 2 places (areas or points) should be shown there.

Figure 10 is small too; its components should be also increased; besides that neither deformed TiO2 nor spherical TiO2 can be seen in fig. 10c

In Figure 13 general structures of coatings are introduced, but not microstructures.

Author Response

We are grateful to the reviewer whose comments are very helpful for us to improve the quality of the current research paper. Please find the individual reply (blue) and the corresponding text highlighted yellow in the manuscript.

In this paper nanostructured TiO2 coatings after depositing onto a stainless steel substrate using the suspension HVOF spray process were described. Different feedstock melting conditions were used and microstructures, hardness, mechanical properties as well as friction and wear behavior were tested. Some remarks can be done:

Key words are incomplete. It would have been better “friction and wear behavior” or “tribological  behavior”.

Reply: Thank for your suggestion and ‘friction and wear behaviour’ has been added into key words to replace ‘friction and wear behaviour’.

What software was used to fulfill XRD phase’s analysis?

Reply: The XRD phase analysis was carried out using software Jade XRD and the standard XRD database. Information has been added in the text in line 106.

The bar graph should be used in fig. 6 instead of the point graph.

Reply: Thanks for the suggestion. We added the bar to the original figure to make them clear visible.

In 3.3 section the specific wear rate is analyzed (see 135 line), but not wear rate.

Reply: Thanks and they are all checked and changed.

Figure 9 is too small; its components should be increased; spectrum 1 and 2 places (areas or points) should be shown there.

Reply: Thanks. We have enlarged the font size as well as redraw the EDS spectra.

Figure 10 is small too; its components should be also increased; besides that neither deformed TiO2 nor spherical TiO2 can be seen in fig. 10c

Reply: Thanks and we have enlarged the font size and components. We forgot to check the labels for deformed TiO2 and spherical TiO2 when we were changing order of the images. They should be in Fig 10f and this has been corrected in the paper. Resolution of images have been improved as well.

In fig. 13 general structures of coatings are introduced, but not microstructures.

Reply: Thanks. Labels have been added in the image to indicate coating and substrate. Text has been added in line 352.

Reviewer 3 Report

In this submission, nanostructured TiO2 coatings deposited by suspension HVOF spraying system, were investigated. The submission is very well organized and written according to scientific-research writing rules.

However, the section Results is a little bit longer than it is needed and in section Discussion, a few unappropriated comparisons or citations are used. For example, the article 23 is not a literature for hardness definition and the paper 29 is not appropriate for comparison and discussion since the plasma sprayed hydroxyapatite coatings are not comparable with HVOF TiO2 coatings at all. Please, reconsider the remaining cited articles, especially in section Discussion.

Furthermore, in section Conclusion, please give us some stronger conclusions (or highlights) about behavior of three different types of coatings. Firstly, why the difference in structure is occurred and secondly, why there is a difference between tribological properties between them. Moreover, please check the following sentences:

ABSTRACT:

“In comparison the coatings composed of mostly fully melted splats (≥ 80%) have the highest coefficient of friction and wear rate and the wear mechanism is dominated by abrasive wear accompanied by the formation of cracks.”

CONCLUSION:

“The coating with maximum agglomerate particles (≤ 12% melted splats) has the lowest wear rate, and the coating with medium melted splats (51%) has the worst performance again wear.”

By the way, please use “against” instead of ”again”.

In addition, the structure of most thermally deposited coatings can be very far from a thermodynamic equilibrium state since it is obtained by high-velocity deposition and cooling processes. Consequently, in investigation of these materials it is good to have repeatability principle fulfilled. Therefore, I would like to suggest to the authors to explain in detail the number of samples used for their valuable investigation.

Did the authors consider using designed experiments (i.e. to have all combinations of spray distance and type of fuel) with the replications for each combination? I believe that, in that way, much more reliable results would be achieved. It would be good to insert some of the pictures of the experimental work (samples, devices for surface engineering).

In many sentences, the authors state that the nanostructured coatings are generally better than conventional. Is it a general conclusion or is obtained by the authors? It seems that the authors did not investigate and compare nanostructured and conventional coatings.  

The authors should explain in detail the rutile and anatase phases.

Is it possible to accomplish the laboratory conditions (especially the cooling of work piece) in real exploitation situations?

Regarding that, explaining the industrial applicability of the research couldbe valuable.

In addition, the scientific value added of the manuscript has to be highlighted. 
Regarding the technical editing of the submission, I would like to suggest to the authors the following:-       

Please do not use the different type of citations (for example, you have Bannier et. al. without year 

then Lima et. al. or Bolleli et. al. without year, but with the used number from the list, … 

Maybe, the use of the numbers is enough; please check the guide for the authors.-       

The title of the Figure 8 should be under the figure.-       Some figures are unclear (for example, 

the Figure 12 a – unclear chemical elements) and should be larger-       The morphology of the samples ….was? 

(instead of were?)-       I suggest to avoid the data about the surface roughness from Table 3, because this table 

is so far from the sentence where it is firstly appeared. 

Author Response

We are grateful to the reviewer whose comments are very helpful for us to improve the quality of the current research paper. Please find the individual reply (blue) and the corresponding text highlighted yellow in the manuscript.

In this submission, nanostructured TiO2 coatings deposited by suspension HVOF spraying system, were investigated. The submission is very well organized and written according to scientific-research writing rules.

Reply: Thank you very much for your good comments and recognition of our work.

However, the section Results is a little bit longer than it is needed and in section Discussion, a few unappropriated comparisons or citations are used. For example, the article 23 is not a literature for hardness definition and the paper 29 is not appropriate for comparison and discussion since the plasma sprayed hydroxyapatite coatings are not comparable with HVOF TiO2 coatings at all. Please, reconsider the remaining cited articles, especially in section Discussion.

Reply: Thanks for your comments. Original ref 23 has been replaced with a more proper one [Herrmann K, Hardness Testing: Principles and Applications, ASM International, 2011]. Original ref 29 has been deleted. Other citations in Discussion section have been checked and original Ref 34 was removed.

Furthermore, in section Conclusion, please give us some stronger conclusions (or highlights) about behavior of three different types of coatings. Firstly, why the difference in structure is occurred and secondly, why there is a difference between tribological properties between them. Moreover, please check the following sentences:

ABSTRACT:

“In comparison the coatings composed of mostly fully melted splats (≥ 80%) have the highest coefficient of friction and wear rate and the wear mechanism is dominated by abrasive wear accompanied by the formation of cracks.”

CONCLUSION:

“The coating with maximum agglomerate particles (≤ 12% melted splats) has the lowest wear rate, and the coating with medium melted splats (51%) has the worst performance again wear.”

By the way, please use “against” instead of ”again”.

Reply: Thank you very much for pointing out our mistakes. Words have been changed. Abstract and conclusions have been updated for a better understanding and highlighted.

In addition, the structure of most thermally deposited coatings can be very far from a thermodynamic equilibrium state since it is obtained by high-velocity deposition and cooling processes. Consequently, in investigation of these materials it is good to have repeatability principle fulfilled. Therefore, I would like to suggest to the authors to explain in detail the number of samples used for their valuable investigation.

Reply: Thank you for your suggestion. Information has been added in line 115 and line 125.

Did the authors consider using designed experiments (i.e. to have all combinations of spray distance and type of fuel) with the replications for each combination? I believe that, in that way, much more reliable results would be achieved. It would be good to insert some of the pictures of the experimental work (samples, devices for surface engineering).

Reply: Thank you for your suggestion. The paper was part of a PhD project where a wider range of parameters were analysed, including spray distances, types of fuel, suspension feed rates. In this paper we selected three typical coatings offering different feedstock melting conditions to study how their microstructures (agglomerated, fully molten, etc) influence their friction and wear properties. A digital image of HVOF Gun with flame has been added in Figure 1b as suggested.

In many sentences, the authors state that the nanostructured coatings are generally better than conventional. Is it a general conclusion or is obtained by the authors? It seems that the authors did not investigate and compare nanostructured and conventional coatings.  

Reply: The nanostructured coatings had indeed better tribological properties than the conventional coatings in our research which was part of the PhD work and will be published later on. This research however was not the focus of this paper.

The authors should explain in detail the rutile and anatase phases.

Reply: Thanks. Both anatase and rutile are tetragonal structures with different c/a ratios of 2.52 and 0.64, and different densities of 3.915 and 4.276 g/cm3 respectively.  Anatase has a relative larger spacing which may accompany with easy slide.  This statement has added in the lines 103-105.

Is it possible to accomplish the laboratory conditions (especially the cooling of work piece) in real exploitation situations? Regarding that, explaining the industrial applicability of the research could be valuable.

Reply: HVOF spraying is a well-developed process for industrial applications and it is flexible with types of feedstocks and sizes of substrates. For this work we used water cooling was used to avoid distortion of substrate from over-heating which is the standard approach in manufacturing thus the current method can be applied to the industry applications.

In addition, the scientific value added of the manuscript has to be highlighted. 
Regarding the technical editing of the submission, I would like to suggest to the authors the following:-       

Please do not use the different type of citations (for example, you have Bannier et. al. without year then Lima et. al. or 

Bolleli et. al. without year, but with the used number from the list, … Maybe, the use of the numbers is enough; please check 

the guide for the authors.-     The title of the Figure 8 should be under the figure.-

Some figures are unclear (for example, the Figure 12 a –unclear chemical elements) and should be larger-       

The morphology of the samples ….was? (instead of were?)-       I suggest to avoid the data about the surface roughness from 

Table 3, because this table is so far from the sentence where it is firstly appeared. 

Reply: Thank you for your comments. Words/citations have been modified accordingly as suggested. The figure caption position (Figure 8) and the font sizes of other figures have been enlarged.

Round 2

Reviewer 3 Report

The comments and suggestions are taken into the consideration and I believe that they were useful.

One more suggestion is to check the references, i.e. some of the journal titles are abbreviated, and majority of them are not. 

Also, check the "Surface and Coatings Technology" and "Surface & Coatings Technology".